# Population Genetic Structure of the Bean Leaf Beetle *Ootheca mutabilis* (Coleoptera: Chrysomelidae) in Uganda

**DOI:** 10.3390/insects13060543

**Published:** 2022-06-14

**Authors:** Dalton Kanyesigye, Vincent Pius Alibu, Wee Tek Tay, Polycarp Nalela, Pamela Paparu, Samuel Olaboro, Stanley Tamusange Nkalubo, Ismail Siraj Kayondo, Gonçalo Silva, Susan E. Seal, Michael Hilary Otim

**Affiliations:** 1National Agricultural Research Organization (NARO), National Crops Resources Research Institute (NaCRRI), Kampala P.O. Box 7084, Uganda; kanyesigyedalton@gmail.com (D.K.); nalelapolycarp@gmail.com (P.N.); pamela.paparu@gmail.com (P.P.); olasamuel94@gmail.com (S.O.); tamusange@gmail.com (S.T.N.); 2College of Veterinary Medicine, Animal Resources and Biosecurity (CoVAB), Makerere University, Kampala P.O. Box 7062, Uganda; 3College of Natural Sciences (CoNAS), Makerere University, Kampala P.O. Box 7062, Uganda; vpalibu@yahoo.com; 4Commonwealth Scientific and Industrial Research Organisation, Canberra, ACT 2601, Australia; weetek.tay@csiro.au; 5International Institute of Tropical Agriculture, PMB 5320, Oyo Rd., Ibadan 20001, Nigeria; kayondowork@gmail.com; 6Natural Resources Institute, University of Greenwich, Central Avenue, Chatham Maritime, Kent ME4 4TB, UK; g.silva@greenwich.ac.uk (G.S.); s.e.seal@greenwich.ac.uk (S.E.S.)

**Keywords:** genetic differentiation, leaf beetle, mitochondrial DNA, microsatellites, haplotype, gene flow, genetic variation

## Abstract

**Simple Summary:**

The bean leaf beetle (*Ootheca mutabilis*) has lately emerged as a major bean pest in Uganda, causing devastating crop losses. Despite its importance, little is known about its population genetic structure. We developed microsatellite DNA markers and combined them with partial mito-chondrial cytochrome oxidase subunit I gene sequences as a marker to examine the spatial pop-ulation genetic structure of 86 *O. mutabilis* samples from 16 populations. We developed a set of five highly polymorphic microsatellite DNA markers. From both types of markers, nearly all the genetic variation occurred within populations and there was no evidence of genetic differentiation in both markers. There was no isolation by distance between geographical and genetic distances for both markers except in one of the agro-ecological zones for mt*COI* data. This information will assist in the design of *O. mutabilis* control strategies.

**Abstract:**

Bean leaf beetle (BLB) (*Ootheca mutabilis*) has emerged as an important bean pest in Uganda, leading to devastating crop losses. There is limited information on the population genetic structure of BLB despite its importance. In this study, novel microsatellite DNA markers and the partial mitochondrial cytochrome oxidase subunit I (mt*COI*) gene sequences were used to analyze the spatial population genetic structure, genetic differentiation and haplotype diversity of 86 *O. mutabilis* samples from 16 (districts) populations. We identified 19,356 simple sequence repeats (SSRs) (mono, di-, tri-, tetra-, penta-, and hexa-nucleotides) of which 81 di, tri and tetra-nucleotides were selected for primer synthesis. Five highly polymorphic SSR markers (4–21 alleles, heterozygosity 0.59–0.84, polymorphic information content (PIC) 50.13–83.14%) were used for this study. Analyses of the 16 *O. mutabilis* populations with these five novel SSRs found nearly all the genetic variation occurring within populations and there was no evidence of genetic differentiation detected for both types of markers. Also, there was no evidence of isolation by distance between geographical and genetic distances for SSR data and mt*COI* data except in one agro-ecological zone for mt*COI* data. Bayesian clustering identified a signature of admixture that suggests genetic contributions from two hypothetical ancestral genetic lineages for both types of markers, and the minimum-spanning haplotype network showed low differentiation in minor haplotypes from the most common haplotype with the most common haplotype occurring in all the 16 districts. A lack of genetic differentiation indicates unrestricted migrations between populations. This information will contribute to the design of BLB control strategies.

## 1. Introduction

In East and Southern Africa, the bean leaf beetle (*Ootheca mutabilis* (Coleoptera: Chrysomelidae)) is an economically important pest of the common bean (*Phaseolus vulgaris* L.) [1]. In the recent past, *Ootheca* species have been classified as field pests of common bean (*P. vulgaris* L.) in Uganda’s Northern and Eastern regions [2,3]. The larvae consume and destroy root tissue, while the adults consume and skeletonize leaves, flowers, and immature pods [4]. During heavy infestations, complete crop losses may occur [1,5]. Bean leaf beetles have also been linked to virus transmission in cowpeas [1,6,7]. *Ootheca mutabilis* is the most common *Ootheca* species (70.3%) in all agro-ecological zones of Uganda [8], despite being reported to be primarily found in lowlands [1,4,8]. However, in Uganda, its abundance and damage to common bean vary depending on location and season [8]. Furthermore, *O. mutabilis* comes in a variety of colours and is frequently confused with other *Ootheca* species such as *O. proteus*, *O. bennigseni*, and *O. orientalis* [9]. Some previously described *Ootheca* species have been re-described and re-assigned to other species, resulting in revisions to the morphological identification records [9]. Presently, the only reliable morphological identification approach is dissection and analysis of male genitalia [9] which, excludes species-level identification of females and other growth stages.

Microsatellite genetic markers, commonly known as simple sequence repeat (SSR) markers, are comprised of tandemly repeated nucleotide motifs of 1–6 base pairs (bp). They have been frequently employed in evolutionary genetic studies [10,11] to infer life history, including mating behavior [10,12,13], mating frequencies [14], gene flow patterns [12], dispersal [15,16], range shifting [15], and host-shifting [17]. The advantages of employing microsatellites in population genetics related research are that they are relatively common, co-dominant, ubiquitous, and substantially polymorphic [11]. However, because of their interaction with mobile elements [18,19], it can be difficult to segregate them into certain insect orders [18,20,21]. Microsatellites have been used in a variety of insects, including the European stag beetle [22] and the red flour beetle [23].

High throughput sequencing (HTS) enables cost-effective and rapid identification of a large number of SSR loci even when only a fraction of the sequencing run is used [24]. The data generated from HTS can be applied in de novo whole-genome sequencing of complex genomes. While exploring this approach, small to large coverage can be performed depending on the researcher’s availability of resources [24].

Recent research on *O. mutabilis* has revealed information on its distribution [4,8], morphological appearance [9], abundance [4,8], and yield losses [8] which, are some of the important information in designing its control strategies. However, there is no available information on its population genetic structure despite being one of the most important considerations in designing long-lasting area-wide pest management strategies. This limits the development of effective area-wide management techniques for these damaging agricultural pests. Understanding BLB population structure, namely if discrete and genetically unique subpopulations exist, is thus necessary for the development of effective control approaches for these destructive pests.

This study aimed to determine the spatial population genetic structure of *O. mutabilis* in Uganda using microsatellite DNA markers and partial mitochondrial DNA cytochrome c oxidase subunit I (mt*COI*). Since there were no existing genetic resources specific to *O. mutabilis* that could be utilized to determine its population genetic structure, we developed SSR markers which we deployed as a genetic tool. Given the desirable attributes of microsatellites, such as codominance, high polymorphism level, and data reproducibility can be instrumental in revealing small-scale resolution of the demographic events [25,26]. Mitochondrial markers are usually helpful in the analysis of earlier phylogeographic events and large-scale patterns of genetic diversity [27,28]. Combining the two markers that experience different modes of inheritance and degrees of polymorphism provides more information that is unattainable through studies where only one of the markers is used. Molecular diagnostic approaches are required to help delineate BLB species status, with the use of partial mitochondrial DNA cytochrome c oxidase subunit I (mt*COI*) sequences being a preferred method globally and successfully employed in our Ugandan laboratory for other agricultural pests (for example, [29,30,31]. Additionally, the mt*COI* partial gene was utilized to barcode the morphologically indistinguishable BLBs prior to population genetic structure analysis, given the unreliability of morphological species identification based on colour patterns alone [9] purposely to avoid misidentification.

## 2. Materials and Methods

### 2.1. Sample Collection

During the 2016–2017 and 2018 rain seasons, we collected BLBs from 17 districts (Figure 1, Table 1 and Table 2) in five bean production agro-ecological zones (Table 1). Adult BLBs were sampled in farmers’ common bean fields. We selected at least one district from each agro-ecological zone, from which two sub-counties were selected. We picked about 10 insects from ten gardens in each sub-county. Each beetle sample was immediately placed individually in a screw-capped 2 mL vial containing 95% ethanol, and the ethanol was replaced at least twice at day intervals to avoid DNA degradation caused by adult beetle secretions. Before further analysis, samples were transported to the National Crops Resources Research Institute (NaCRRI), Namulonge in Wakiso district, and kept in boxes at room temperature. During sample collection, a GPS coordinate was taken for each garden from which samples were picked, and each GPS point was regarded as one sample since only one sample (BLB) was considered for every GPS point where BLBs were recovered. From every farmer’s field where BLBs were found, at least one BLB was picked. In one of the districts (Lwengo) selected for sample collection in the Central Wooden Savannah agro-ecological zone, BLBs were not recovered at the time of collection (Table 2). This, therefore, reduced the number of districts as well as the number of samples in this agro-ecological zone.

### 2.2. Identification of Bean Leaf Beetles

Representatives of BLB samples were identified morphologically at the Universitätkoblenz-Landau Institut für Integrierte Naturwissenschaften Abteilung Biologie Universitätsstraße 156,070 Koblenz, Germany, purposely to establish the identity of *Ootheca* species which information would help us in mass collecting of the true *Ootheca* species for laboratory analysis. Before the collection of the samples for laboratory analysis, various leaf beetles collected from the bean plants were sorted to pick representatives of each morphotype which were considered for morphological analysis. As described by [9], the BLB samples studied for the population genetic structure were selected for DNA analysis based on colour patterns of the elytra, head, thorax, abdomen, and legs. In this regard, we selected 99 samples based on their appearance as follows: M1 (*O. mutabilis* with elytra upper half black and lower half brownish) (21 beetles), M2 (*O. mutabilis* with black elytra) (39 beetles), and M3 (*O. mutabilis* with brownish elytra) (39 beetles) (Figure 2). This total included the samples found to be *O. proteus* before DNA barcoding as they could not be at all distinguished from *O. mutabilis*. All the BLBs with different colour appearances as explained above and shown in Figure 2, as well as reported by [9], were included in the analysis. This colour distinction was made to examine potential genetic variations between *O. mutabilis* morphotypes. At least, every district from which BLB samples were recovered, was considered as well as some of the samples collected. The number of beetles per district considered for analysis was different because after DNA barcoding, non-*O. mutabilis* samples were removed (Hoima and Nakasongola) (Table 2). Also, because we wanted to consider different morphotypes, in some districts, samples were increased so that different morphotypes would be included since some districts had one morphotype and others had two or all three morphotypes. Also, during sample selection in the laboratory, one sample was considered per GPS point.

### 2.3. DNA Isolation and Quantification

For DNA isolation, we used the Qiagen DNeasy Blood & Tissue Kit (Hilden, Germany) in accordance with the manufacturer’s instructions. Following isolation, each DNA sample was quantified using an Agilent Technologies NanoDrop 2000 spectrophotometer (Waldbronn, Germany) and the quality was confirmed using 1% agarose gel electrophoresis.

To confirm their identity, all samples selected based on colour were barcoded using the mt*COI* partial gene primers. The PCR primers used were BLB-LCO: 5′-GGTCAACAAATCATAAAGATATTGG-3′ and BLB-HCO: 5′-TAAACTTCAGGGTGACCAAAAAATCA-3′, which amplify a 710 bp fragment (M. Otim unpublished). Each reaction was carried out in a 25 µL volume comprising 1 µL of template DNA, 1 µL of 10 pmol/µL primer, 2.5 µL of 10X DreamTaq Green buffer, 0.5 µL of dNTP (10 mM), 0.25 µL (1.25 units) of Taq DNA polymerase (5 U/µL), 2.5 µL 5% tween20, and 16.25 µL of nuclease-free water. The PCR conditions were as follows: 1 cycle of denaturation at 95 °C for 2 min, 35 cycles of 20 s at 95 °C, 30 s at 52 °C annealing temperature, 1 min at 72 °C, and a final extension cycle at 72 °C for 10 min, after which reactions were held at 4 °C. All samples were tested for amplification success on 1.3% *w*/*v* agarose in TAE buffer and stained with ethidium bromide as stated above. Sequences generated from mt*COI* PCR products were processed with Pregap4 and Gap4 [32] and compared to NCBI sequences of *O. mutabilis* (KY574530.1, KY574526.1, KY574527.1) and *O. proteus* (KY574525.1, KY574524.1, KY574523.1, KY574522.1).

### 2.4. Genome Sequencing, Quality Check and Raw Read Assembly

GENEWIZ (www.genewiz.com was contracted to carry out high throughput sequencing (HTS). DNA was isolated from three ethanol-preserved BLBs denoted by the letters M1, M2, and M3 (Figure 2). The Illumina HiSeq 2500 system was used to construct and sequence whole-genome DNA libraries, with an insert size of 300 to 400 bp and 2 × 150 paired-end reads. Fast QC v0.11.7 [33] was used to check the quality of raw reads before processing them for de novo assembly. Following QC, raw reads were processed, with adaptor sequences trimmed, duplicate sequences eliminated, and the sequences assembled de novo. All raw read processing and de novo assembly were performed in Geneious v10.2 [34] using the default settings (i.e., allow gaps—maximum gaps per read 20%, ignore words repeated more than 1000 times, do not merge variants with coverage over approximately 6, merge homopolymer variants).

### 2.5. Microsatellite Prediction, Primer Design and Blast Search of Microsatellite Sequences in GenBank

WebSat [35] was used to identify microsatellites in the assembled contigs, and Primer3 [36] was used to design primers. The primer design parameters included a primer size range of 18 nucleotides at the lowest, 22 at the optimum, and 27 at the maximum. Primer Tm was 57 °C at minimum, 60 °C at optimum, and 68 °C at maximum. The primer GC% ranged from 40 to 80. The product size range was 100–400 bp. After designing the primers, 81 desalted primer pairs were ordered from Macrogen Europe (dna.macrogen.com). Primers were designed for all di-nucleotide, tri-nucleotide, and tetra-nucleotide microsatellite loci identified. Microsatellite sequences were compared with microsatellite sequences in the NCBI GenBank using BLASTX and BLASTN [37] to find out if they shared similarities with sequences from other insects and putative retrotransposable elements (e.g., [18]).

### 2.6. Microsatellite DNA Marker PCR Optimization, Polymorphism Testing, Primer Labelling and Fragment Analysis

Each microsatellite primer pair was optimized for amplification prior to being evaluated for polymorphism on eight *O. mutabilis* samples (at least one sample from every population). Each locus was amplified in a 12.5 µL PCR reaction containing 0.5 µL of 50 ng DNA template, 0.5 µL of 10 pmol/L primer, 1.25 µL of 10X DreamTaq green buffer, 0.25 µL of DreamTaq dNTP (10 mM), 0.125 µL of DreamTaq DNA polymerase (5 U/µL), 1.25 µL% Tween20, and 8.125 µL of nuclease-free water. The PCR conditions were as follows: initial denaturation (4 min 94 °C), 35 amplification cycles (20 s 94 °C, 30 s annealing temperature, 45 s 72 °C), and a final extension (10 min 72 °C). Then, the reactions were held at 4 °C. The amplicon was run on a 3% *w*/*v* agarose in 1X TAE buffer at first, then on 6% acrylamide gels for 6 h at 120 V with a 100 bp DNA ladder in 1X TAE buffer. Polyacrylamide gels were stained for at least 20 min in an ethidium bromide (0.5 g/mL) solution before being de-stained in distilled water. The U-genius gel documentation system was used to visualize fragment sizes (www.syngene.com. Each locus was tested at least twice for reproducibility. Following primer optimization, five primers were selected based on polymorphism on polyacrylamide gels. These primers were ordered for synthesis from Macrogen Europe (dna.macrogen-europe.com). Each forward primer was labelled with 6-FAM or HEX fluorescent dyes (dna.macrogen-europe.com) (Table 3). The PCR reactions were carried out in single reactions, and then the 6-FAM and HEX PCR products with similar amplicon product size ranges were pooled and run as multiplex. The Applied Biosystems 3730XL DNA Analyzer was used to analyze fragments (outsourced to Macrogen Europe).

### 2.7. Genotyping and Data Scoring

Genotyping was performed using GeneMarker v2.6.3 [38], and alleles were scored based on their size. MICROCHECKER [39] was used to assess the accuracy of allele scoring. MolKin v3.0 [40] was used to determine polymorphic information content (PIC), observed and expected heterozygosities (*H_o_*, *H_e_*).

### 2.8. Population Genetic Structure and Differentiation Analysis

During data analysis, different districts were treated as different populations leading to a total of 16 populations as detailed (Table 2). The elimination of morphologically indistinguishable samples from the analysis reduced the originally selected samples from 99 to 86.

DnaSP v6 was used to compute mt*COI* partial gene diversity components such as the number of haplotypes, haplotype diversity, nucleotide diversity, and polymorphic site estimates [41].

We investigated population genetic structure by AMOVA that partitions total variance into covariance components. It then verifies the hierarchical or non-hierarchical variation distribution (i.e., among populations (fixation index (*F*_ST_), among populations within groups (*F*_SC_) and among groups (*F*_CT_)) [42]. Analysis of molecular variance (AMOVA) for both sets of data (each data set analysed independently) was performed in Arlequin version 3.5 using 1000 permutations, all at a 0.05 significant level. Analysis of molecular variance (AMOVA) was performed using both non-hierarchical (i.e., all populations in one group) and hierarchical designs (i.e., populations subdivided into five agro-ecological zones as detailed in Table 2). The spatial analysis of molecular variance for both types of markers was calculated using SAMOVA 2.0 software [43] to obtain inferences about the population groups. The SAMOVA for both types of markers were analyzed using 1000 simulated annealing processes by varying K (number of groups) from 2 to 10. The best K value was selected according to when *F*_CT_ reached a plateau. The best K grouping was used to calculate AMOVA in Arlequin to estimate genetic differentiation.

Geographic structures of *O. mutabilis* populations for both types of markers were investigated using the Bayesian approach implemented in STRUCTURE v2.3.4 [44]. STRUCTURE uses a coalescent genetic technique to group similar multilocus genotypes into inferred ancestral genetic clusters (K), regardless of an individual’s geographical origin. Using the admixture model, we conducted ten independent runs for each value of K ranging from 1 to 5. Each run consisted of a burn-in of 50,000 steps followed by 100,000 Markov-Chain Monte Carlo (MCMC) repetitions. For each potential value of K, ten replicates were used. The mt*COI* sequences were first processed so that only haplotypes are used in the inference as explained in the STRUCTURE user manual [44]. The LOCPRIOR command was used to perform the STRUCTURE runs, and the genotypes defined were based on the geographic location of the *O. mutabilis* samples. The true value of K was estimated using the program STRUCTURE HARVESTER [45], as described in [46]. CLUMPAK [47] was used to visualize the structural results.

The mt*COI* haplotype network was inferred using the minimum spanning network approach described in POPART [48] using 2000 bootstraps based on sequence alignment exported from DnaSP v6 as a haplotype nexus file.

### 2.9. Isolation by Distance (IBD) Analysis

Evidence of IBD for both types of markers was tested using the Mantel test [49] with 9999 permutations in GenAIEx v6.503 [50]. Isolation by distance was calculated for different agro-ecological zones as detailed in Table 2.

## 3. Results

### 3.1. Quality Check of NGS Sequences, De Novo Assembly, SSR Prediction and Primer Design

A total of 272,853,156 raw reads were generated, with equal forward and reverse reads. The length of all sequences was 151 bp, with a GC content of 34%. QC results included: basic statistics passed; per base sequence quality passed; per tile sequence quality passed; per sequence quality scores passed; per base sequence content passed; and, sequence length distribution passed. There was no over-representation of the sequences. The assembly of all sequences in the selected part of the reads resulted in a total of 282,696 contigs. From the assembled contigs, a total of 19,356 SSR were identified, including (i) mononucleotides (14,629; 75.6%), (ii) di-nucleotides (2780; 14.4%), (iii) tri-nucleotides (1288; 6.7%), (iv) tetra-nucleotides (352; 1.8%), (v) penta-nucleotides (258; 1.3%), and (vi) hexa-nucleotides (49; 0.3%).

### 3.2. Microsatellite PCR Optimization and Polymorphism Testing

Sixty-five of the 81 loci analyzed showed multiple fragments on agarose gels and were therefore removed from the analysis. On agarose gels, a total of 16 loci were observed as a single band, and they also appeared polymorphic on 6% acrylamide gels. Six of the 16 polymorphic loci were eliminated due to inadequate PCR amplification (i.e., fuzzy bands in some samples, and failed PCR amplification in others). Two of the ten loci had primers that overlapped the (GT) SSR units and were thus eliminated. We eliminated three of the remaining eight loci due to low repeatability. As a result, five loci were chosen for genotyping (representing a 6% success rate) and labelled with fluorescent dyes (Table 3). The number of SSR alleles ranged from 4 to 21, with an average of 11.6. (Table 3).

### 3.3. Fragment Analysis and Allele Scoring

A locus was judged to have low polymorphism if the PIC was less than 25% and high polymorphism if the PIC was greater than 50% [51]. Locus BLB2_om66 had the lowest PIC of 50.1% while BLB2_om33 had the highest PIC of 83.1% (Table 4). The five loci employed in this investigation were, therefore, all were considered to be of high polymorphism, with an average PIC of 69.1%.

At the population level, the average PIC based on all loci across all populations was 53.59% (Table 5), while the population average heterozygosity for all loci was 0.73, with BLB2_om33 having the highest heterozygosity of 0.84 and BLB2_om66 having the lowest heterozygosity of 0.59. (Table 4). Observed heterozygosity was higher than expected heterozygosity in all populations, ranging from 0.75–0.84, with an average observed heterozygosity of 0.80 (Table 5). The loci BLB2_om17 and BLB2_om32 had an excess of homozygosity, which could indicate the potential presence of null alleles or allele drop-out.

### 3.4. Population Genetic Structure, Differentiation and Gene Flow

The 86 *O. mutabilis* mt*COI* partial gene sequences were analyzed, and 21 segregating sites (S) with an average of 0.827 nucleotide differences were found (Kt). We found 20 haplotypes with an estimated haplotype diversity (H, also known as gene diversity, a measure of the probability that two random alleles are different [52]) of 0.51 and a low nucleotide diversity Pi (π, i.e., the average number of nucleotide differences per site in pairwise DNA sequence comparison [52]) of 0.00127. We found moderate haplotype diversity among the 20 haplotypes (GenBank: MW278873-MW278892) (Figure 3).

The minimum spanning haplotype network (Figure 3a) revealed one major haplotype (haplotype 1) that was detected in all five agro-ecologies (A, B, C, D, and E) and 69.77% of the individuals (*n* = 60). The second most prevalent haplotype was haplotype 14, which was present in 5.81% (*n* = 5). Three haplotypes (6, 7 and 18) had 2.30% with two individuals each. The remaining 15 haplotypes each had *n* = 1. (i.e., 1.16% each) (Figure 3a). Haplotypes 6, 7 and 18 with the third-highest frequency (2.30% each) varied in agro-ecologies although most of the individuals (*n* = 4) belonged to agro-ecological zone A (Figure 3a).

In the 16 populations, the number of segregating sites ranged from 0 to 6 with Lira, Amuru, and Bulisa having the highest number of haplotypes, haplotype diversity ranged from 0 to 0.9 among the districts with Lira district having the highest and nucleotide diversity was low in all the populations (districts) (Table 6).

Another minimum spanning haplotype network (Figure 3b) for the samples from different districts was inferred which resulted in all the individuals from all different districts sharing haplotype 1. In both haplotype networks, all other haplotypes originate from haplotype 1 (Figure 3b) indicating low differentiation from the most common haplotype.

The neighbour-joining phylogenetic tree constructed (with one sample (representative) from each haplotype) clearly clustered together all the *O. mutabilis* haplotypes with the *O. mutabilis* reference sequences (KY574530.1, KY574526.1, KY574527.1) with high node support (100%), but with distinct intra-species differences (Figure 4). The *O. proteus* samples that were morphologically indistinguishable from *O. mutabilis* were clearly clustered together, with *O. proteus* reference sequences (KY574525.1, KY574524.1, KY574523.1, KY574522.1) likewise with intra-species differences (Figure 4).

Hierarchical analysis of molecular variance (AMOVA) for both types of markers resulted in all the genetic variation (100%) occurring within the populations. Non-hierarchical AMOVA for microsatellites had the highest genetic variation occurring within the populations and the lower genetic variation occurred among the populations (Table 7(a1)) except for non-hierarchical AMOVA for mt*COI* where all the genetic variation occurred within the populations (Table 7(b1)). For SSRs, the fixation indices among groups, among populations within groups, and within populations as calculated with hierarchical AMOVA were *F*_CT_ = 0.02073 (*p* = 0.05), *F*_SC_ = −0.01523 (*p* = 0.05), and *F*_ST_ = 0.00582 (*p* = 0.05) (Table 7(a2)) indicating that there is low differentiation and relatively high gene flow. The same result trend was also detected in the mt*COI* marker for hierarchical AMOVA (Table 7(b2)).

From the analysis by SAMOVA, for both microsatellites and mt*COI* data, the *F*_CT_ values reached a plateau at K = 2, *F*_CT_ = 0.4, *p* = 0.058 and 0.094, *p* = 0.05474 for SSRs and mt*COI* respectively (Table 8(a1) and 8(a2) respectively) by SAMOVA. The suggested structure by SAMOVA for SSR data resulted in the highest genetic variation occurring within populations and for mt*COI* data, all the genetic variation occurred within populations (Table 8(a3) and 8(a4) respectively).

Based on the results from AMOVA for both markers, *F*_ST_ (0.00069) (SSR markers) and *F*_ST_ (−0.01605) (mt*COI*), genetic differentiation was low.

Based on both SSR and mt*COI* partial gene markers, STRUCTURE analysis identified K = 2 as the likely optimal number of ancestral genetic clusters, with all individuals categorized as one dominant cluster (blue colour) and different degrees of genetic contributions (orange colour) from a second minor genetic cluster (Figure 5a,b).

### 3.5. Isolation by Distance (IBD)

Isolation by distance (IBD) was calculated for each agro-ecological zone independently. During this inference, agro-ecological zones were regarded as populations as detailed in Table 2. There was no evidence of IBD (SSR data) detected (Mantel test for agro-ecological zones A1 to E1 respectively: ((A`1) r = −0.061, *p* = 0.222; (B1) r = 0.107, *p* = 0.205; (C1) r = 0.204, *p* = 0.332; (D1) r = −0.038, *p* = 0.373; (E1) r = −0.073, *p* = 30.363 (Figure 6)). Isolation by distance was also calculated for mt*COI* partial gene data for each agro-ecological zone independently. There was no evidence of IBD detected also ((A2) r = −0.044, *p* = 0.342; (B2) r = −0.128, *p* = 0.685; (C2) r = −0.050, *p* = 0.755; (E2) r = −0.153, *p* = 0.273) except in one of the populations ((D2) agro-ecological zones ((D). Agro-ecological zone D2) ((r = 0.161, *p* = 0.073).

## 4. Discussion

Prior to designing robust and effective pest management strategies, information about the level of individuals’ interaction among locations [53], in addition to pests’ life cycles and their predators [54], need to be availed. The former (pests’ interaction among locations) can be availed through the analysis of the population genetic structure of the pest in question [55].

Our study is the first to report the spatial population genetic structure of *O. mutabilis* in Uganda. Based on both types of markers, we found that the highest genetic variation occurred within the populations, low among groups and none among groups within populations. While sample sizes could account for the low genetic variation detected among groups and none were detected among groups within-population level, this was likely also an indication that there was intermixing or interbreeding between individuals from different populations. This aspect can be attributed to the capabilities for migration of our studied species as beetles are reported to fly short distances, although such trivial flights can also lead to the covering of longer distances [55,56]. Our results concur with a previous report which showed that highly dispersing phytophagous arthropods are characterized by homogenizing effects due to gene flow over distant localities [57]. Another leaf beetle, *Cerotoma trifurcata*, from the mid-western United States [55] exhibited similar high variability within samples, while the lowest degree of variation was among populations, indicating substantial levels of gene flow. Low levels of variation among populations can be attributed to the presence of few or no geographical barriers that impede gene flow. The absence of geographical barriers among *C. trifucarta* communities in the mid-western United States facilitated dispersal and reduced geographical fragmentation and genetic differentiation [55].

There was no evidence of genetic differentiation for *O. mutabilis* for both microsatellite and mt*COI* partial gene molecular markers indicated by insignificant *F*_ST_ values. This suggests that migration occurs across *O. mutabilis* populations, particularly those separated by large geographic distances. Local and long-distance migrations of *O. mutabilis* may be aided by the presence of other host plants and staggered bean planting. Beans are grown at different times of the year in the study area as one of the cultural practices to manage BLBs [8], and this practice has the potential to drive BLB migration between gardens. However, no information is available to confirm whether the presence of beans in the gardens attracts BLBs feeding on beans from distant gardens. It is also possible that BLBs prefer certain bean varieties over others, causing them to migrate between gardens. As a result, it will also be necessary to investigate if volatiles in bean plants can attract BLBs and encourage their migration as it has been reported to occur in *Spodoptera littoralis* caterpillars while feeding on maize [58].

Bean leaf beetles oviposit in the soil, and the eggs hatch into larvae, pupae, and finally adults [5,8]. However, it is unknown whether these eggs can be carried from one garden to another via farm tools such as pangas, hoes, gum boots, and so on.

Our study areas were limited only to Uganda, however, BLBs have been reported to occur elsewhere including in countries from East Africa (e.g., Tanzania) and West Africa (e.g., Senegal) [9]. It would be a greater opportunity to understand their genetic status as the East African Rift Valleys have been shown to support population substructure and/or early speciation in both invertebrates (e.g., [59,60,61]) and vertebrates [1,47]. It remains to be investigated whether the geographical distribution of *O. mutabilis*, which, crossed the Rift Valley, may have resulted in comparable incipient speciation as observed in the cassava whitefly *Bemisia* ‘SSA1’ species using a whole-genome analysis technique [59]. Mitochondrial DNA markers, such as the partial *COI* gene, are ineffective for distinguishing between closely related subspecies [59,62,63].

STRUCTURE analyses from SSR markers revealed genetic mixing from two ancestral genetic lineages in our *O. mutabilis* samples. Based on limited nuclear (SSR) markers and the maternally inherited mt*COI* marker, our findings showed that the genetic composition of the current populations could be explained as the outcome of genetic contributions from two hypothetical ancestral *O. mutabilis* genetic clusters. This study detected no evidence of genetic differentiation indicating that the Ugandan population likely represents a single panmictic population. To gain a better knowledge of the evolutionary genetics and landscape adaptability of this major agricultural coleopteran pest complex, an in-depth population structure study based on whole-genome sequencing would be desirable.

The inability to distinguish species based on physical characteristics is a hindrance to effective pest management. Both *O. mutabilis* and *O. proteus* are important bean pest species that are morphologically indistinguishable and can only be separated by dissection and study of the male genitalia [9], a process that does not distinguish between female species. The mt*COI* gene can help identify morphologically challenging (e.g., [31,64]) and cryptic species (e.g., [65]) including Coleoptera (e.g., [66]). Molecular identification using the mt*COI* partial gene, as described in this study, successfully distinguished between *O. mutabilis* and *O. proteus* that could not be distinguished by colour appearances alone during laboratory sample selection and should be used in future evolutionary genetic studies of *Ootheca* species.

Our IBD analysis results also showed that *O. mutabilis* has had little or no barriers to free migration in all populations. In research conducted by Krell et al. [56], a single *C. trifurcata* beetle travelled 4.9 km. These trivial flights in search of mates, oviposition sites, and food [56] may contribute to greater regional coverage over time, resulting in minimal genetic differentiation and substantial gene flow [55]. Behavioural research findings may aid in identifying features that contribute to analogous IBD and gene flow findings in our target species.

## 5. Conclusions

This study found that almost all the genetic variation in our target species, *O. mutabilis*, occurred within populations which, we attributed to dispersal that facilitated genetic mixing between populations. *Ootheca mutabilis* were divided into two population groups by SAMOVA at K = 2. When inferred, the structure suggested by SAMOVA resulted in almost all the genetic variation occurring within the populations. There was no evidence of genetic differentiation as seen from insignificant *F*_ST_ and we attributed it to gene flow between different *O. mutabilis* populations. On inferring the Bayesian clustering, STRUCTURE at K = 2 categorized all the samples as one dominant cluster (admixture) and different degrees of genetic contributions from a second minor genetic cluster. Therefore, the Ugandan populations of *O. mutabilis* likely represent a single panmictic population with genetic contributions from two ancestral lineages. Our study provides a baseline for future evolutionary and functional genomic studies to generate a better understanding of host-plant adaptation, insecticide resistance management, and the development of integrated pest management control measures for this important pest. Future research on *O. mutabilis* should increase coverage of samples from other African countries via a whole-genome sequencing approach.

## Figures and Tables

**Figure 1 insects-13-00543-f001:**
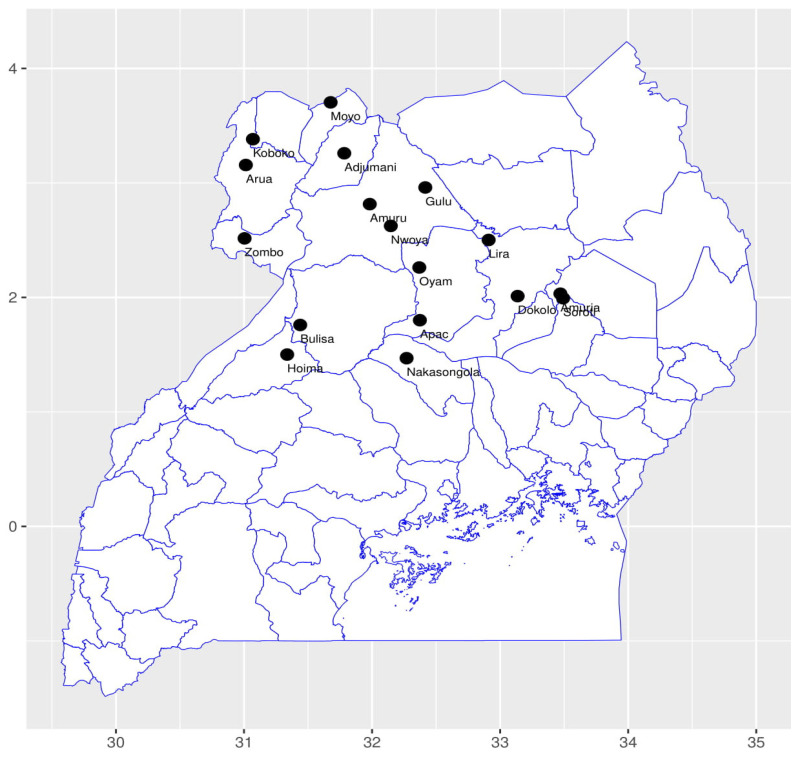
Districts of Uganda collection locations of bean leaf beetle samples used in the study.

**Figure 2 insects-13-00543-f002:**
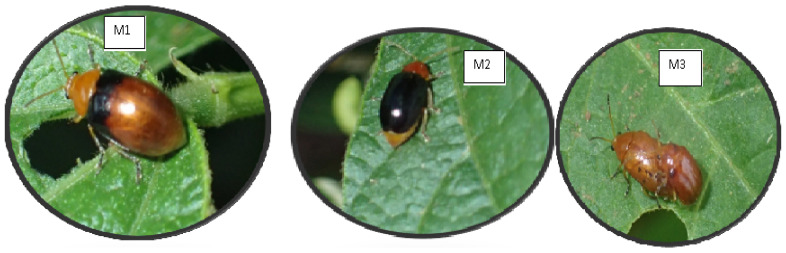
Different colourmorphs of *Ootheca mutabilis.* M1: *O. mutabilis* with elytra upper half black and lower half brownish; M2: *O**. mutabilis* with black elytra; M3: *O. mutabilis* with brown elytra.

**Figure 3 insects-13-00543-f003:**
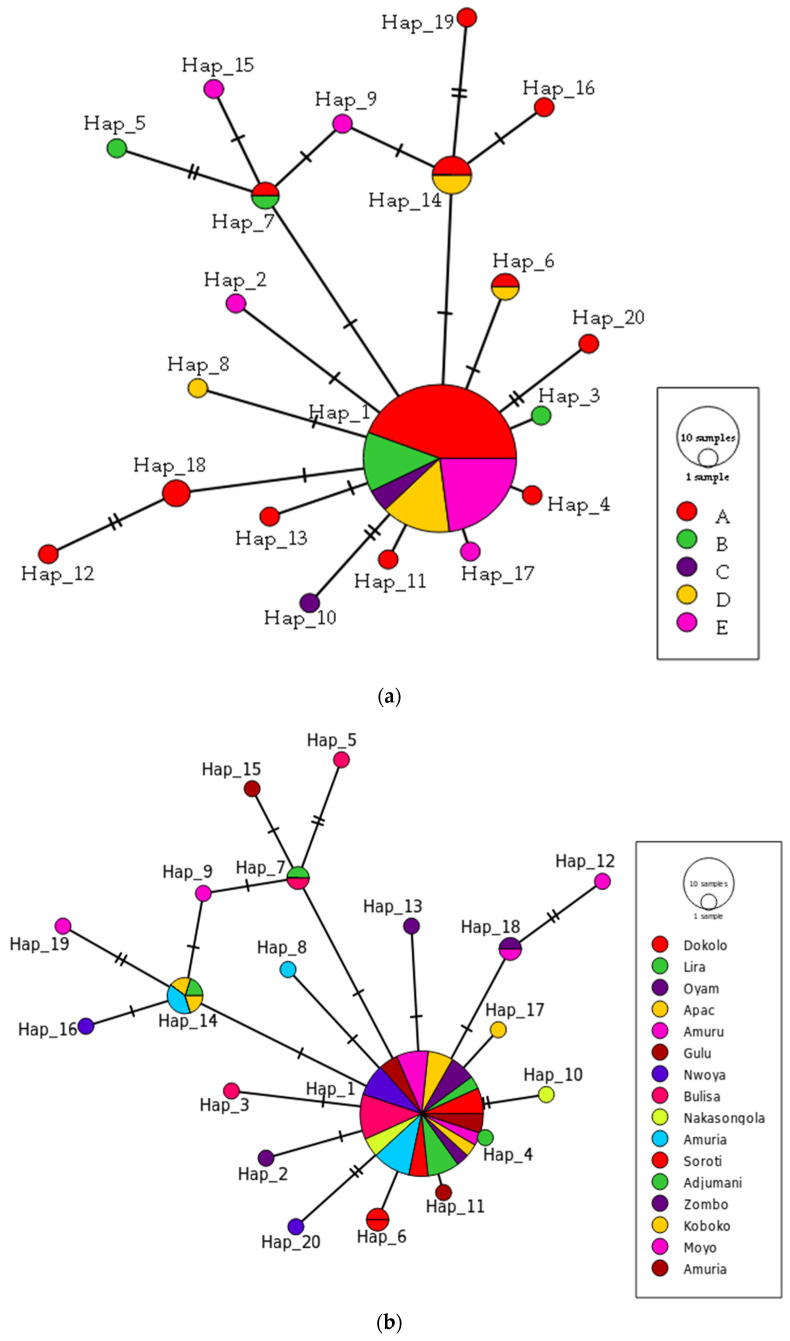
Minimum spanning haplotype network showing evolutionary relationships among haplotypes in different agro-ecological zones (**a**), and different districts (**b**). Each small black line along a connecting line represents a change of one base pair. Haplotypes are colour coded according to the population. In haplotype network (**a**) (red represents population (A) Northern moist farmlands, green represents population (B) Western mid-altitude farmlands, purple represents population (C) Central wooden savannah, yellow represents population (D) Southern and Eastern Lake Kyoga basin and pink represents population (E) North-western farmlands). Circle sizes correspond to the haplotype numbers.

**Figure 4 insects-13-00543-f004:**
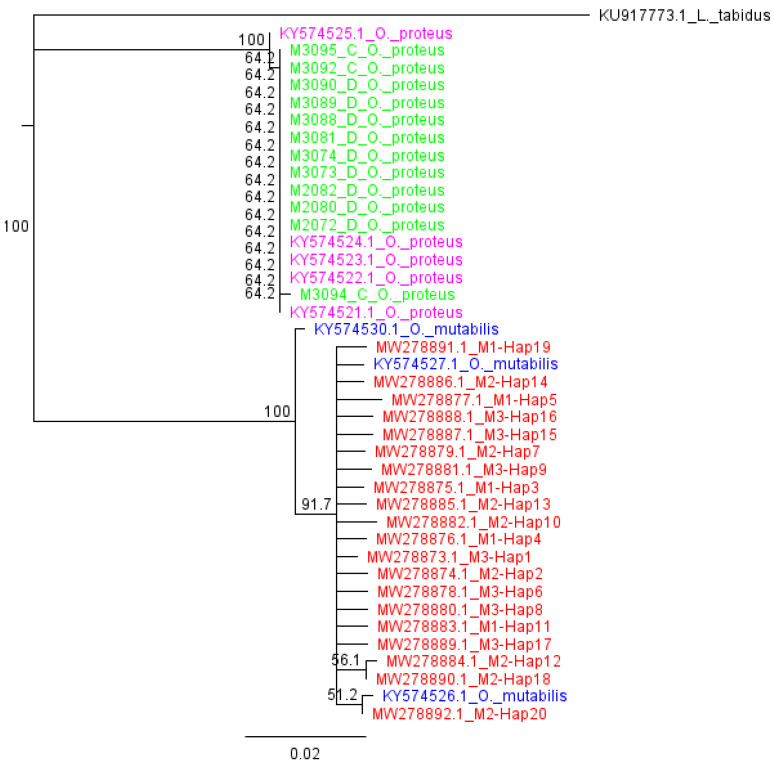
Neighbour Joining phylogenetic tree of the 20 *O. mutabilis* haplotypes (red) found in the study, and reference sequences downloaded from NCBI as follows: *O. mutabilis* (blue), *O. proteus* (pink), *O. proteus* (green, morphologically similar samples to *O. mutabilis* separated after DNA-barcoding) and the outgroup *Longitarsus tabidus*.

**Figure 5 insects-13-00543-f005:**
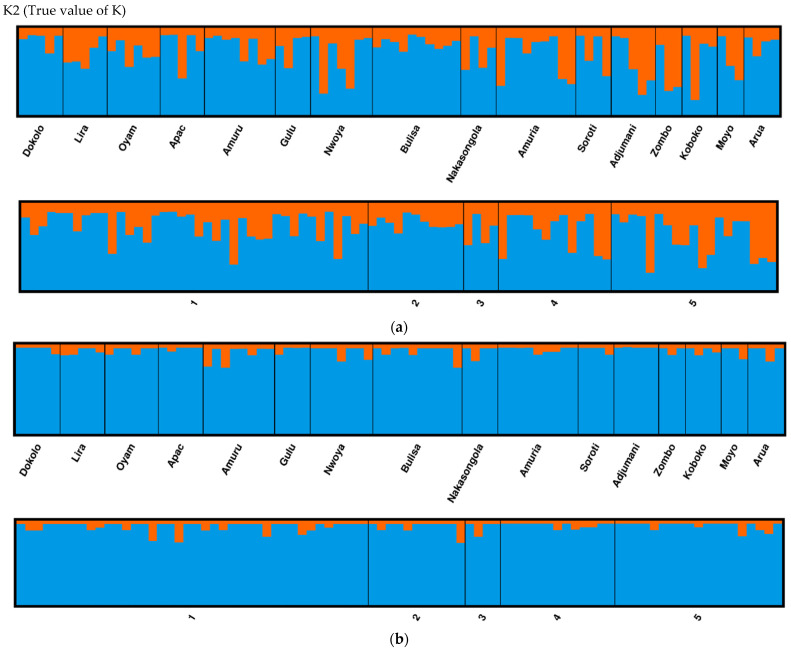
(**a**). Population structure across five and sixteen populations agro-ecological zones and districts respectively of *Ootheca mutabilis* obtained using structure (for SSR data and (**b**)) for mt*CO1* partial gene sequences both at K = 2. Vertical bars represent individuals. Numbers on the horizontal axis represent agro-ecological zones; (1) Northern moist farmlands (A), (2) Western mid-altitude farmlands (B), (3) Central wooden savannah (C), (4) Southern and Eastern Lake Kyoga basin (D), and (5) North-western farmlands (E).

**Figure 6 insects-13-00543-f006:**
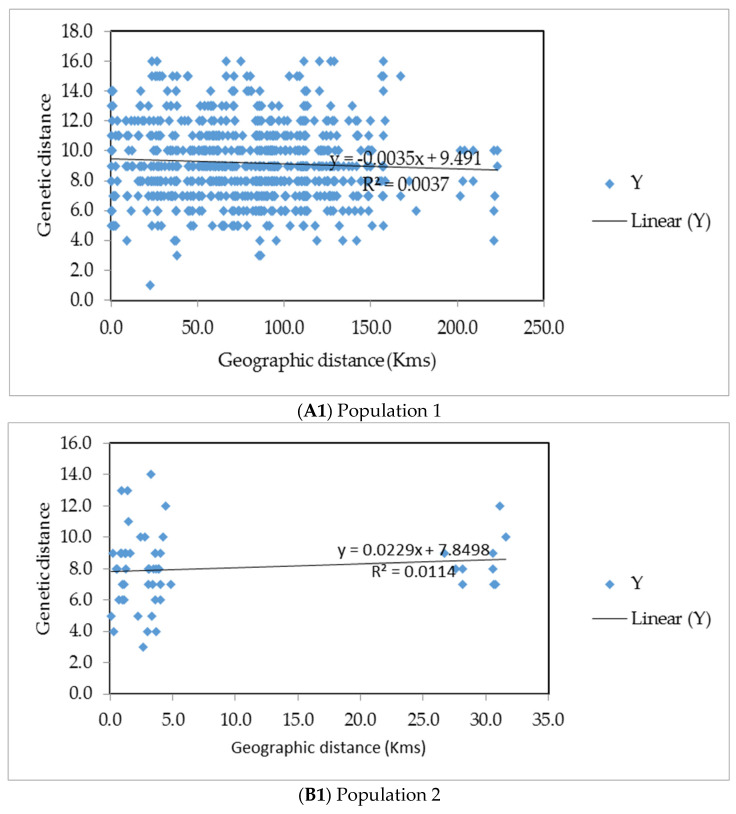
Correlation between geographical and genetic distances of *Ootheca mutabilis* samples for each independent agro-ecological zone. (**A1**,**A2**) Northern moist farmlands (Agro-ecological zone A), (**B1**,**B2**) Western mid-altitude farmlands (Agro-ecological zone B), (**C1**,**C2**) Central wooden savannah (Agro-ecological zone C), (**D1**,**D2**) Southern and Eastern Lake Kyoga basin (Agro-ecological zone D) and (**E1**,**E2**) North-western farmlands (Agro-ecological zone E)). (**A1**–**E1**) represent SSR data and (**A2**–**E2**) represent mt*COI* partial gene data.

**Table 1 insects-13-00543-t001:** Ugandan districts from which the samples were picked, GPS coordinates for each district and the total number of samples picked per district.

District	Latitude	Longitude	Collectors	Samples Collected
Dokolo	2.01148	33.1367	Charles. H, Dalton. K, Sam. O, Sekandi. W	58
Lira	2.50196	32.91012	Charles. H, Dalton. K, Sam. O, Sekandi. W	41
Oyam	2.3559	32.60652	Charles. H, Dalton. K, Sam. O, Sekandi. W	30
Apac	1.88446	32.37174	Charles. H, Dalton. K, Sam. O, Sekandi. W	35
Amuru	2.81492	31.98196	Charles. H, Dalton. K, Sam. O, Sekandi. W	45
Gulu	2.96032	32.41548	Charles. H, Dalton. K, Sam. O, Sekandi. W	36
Nwoya	2.62473	32.14631	Charles. H, Dalton. K, Sam. O, Sekandi. W	42
Bulisa	1.76197	31.43002	Charles. H, Dalton. K, Sam. O, Sekandi. W	45
Hoima	1.50097	31.33689	Charles. H, Dalton. K, Sam. O, Sekandi. W	20
Nakasongola	1.46927	32.2695	Charles. H, Dalton. K, Sam. O, Sekandi. W	25
Amuria	2.0574	33.50213	Charles. H, Dalton. K, Sam. O, Sekandi. W	37
Soroti	5.37120	21.94900	Charles. H, Dalton. K, Sam. O, Sekandi. W	22
Adjumani	3.25633	33.7836	Charles. H, Dalton. K, Sam. O, Sekandi. W	30
Zombo	2.51592	31.00431	Charles. H, Dalton. K, Sam. O, Sekandi. W	20
Koboko	3.38206	31.06935	Charles. H, Dalton. K, Sam. O, Sekandi. W	21
Moyo	3.70361	31.67643	Charles. H, Dalton. K, Sam. O, Sekandi. W	24
Arua	3.15309	31.01043	Charles. H, Dalton. K, Sam. O, Sekandi. W	23

**Table 2 insects-13-00543-t002:** Agro-ecological zones, number of samples of *O. mutabilis* and *O. proteus* found per agro-ecological zone after DNA barcoding with mt*COI* partial gene and Ugandan districts belonging to particular agro-ecological zones where bean leaf beetle samples were collected.

Population Code	Agro-Ecological Zone	Number of *O. mutabilis* Samples Analyzed		Districts
		*O. mutabilis*	*O. proteus*	
**A**	Northern moist farmlands	5	0	Dokolo
		5	0	Lira
		6	0	Oyam
		5	0	Apac
		8	0	Amuru
		4	0	Gulu
		7	0	Nwoya
**B**	Western mid-altitude farmlands	10	0	Bulisa
		1	9	Hoima
**C**	Central wooden savannah	4	3	Nakasongola
		0	0	Lwengo
**D**	Southern and Eastern Lake Kyoga basin	9	0	Amuria
		4	0	Soroti
**E**	North-western farmlands	5	0	Adjumani
		3	0	Zombo
		4	0	Koboko
		3	0	Moyo
		4	0	Arua

**Table 3 insects-13-00543-t003:** Characteristics of the five microsatellite loci developed and used for the *Ootheca mutabilis* population genetic study. NA (Number of alleles).

Locus Name	Motif	Size (bp)	Primer Sequence 5′ 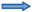 3′	GenBank Accession Number	NA	Tm (°C)	Fluorescent Label
BLB2_om1	(GAT)_2_(CAA)_11_	343–365	F: TCAACTACCACCATCACAAACCR: CAATGTGGAGCAACTACGTCAT	MT074093	9	58	5′6-FAM
BLB2_om17	(CTT)_10_	368–396	F: CCAATCCGCTTCTCTATATCCAR: GGAGCAATGTTATGCCTGATTT	MT074094	16	57	5′6-FAM
BLB2_om32	(GACG)_6_	160–195	F: CATATAGCGAAAACCCGAAATCR:AGAAGTACAAGTATGGCCCGAA	MT074096	21	58	5′6-FAM
BLB2_om33	(ACA)_5_.(ACG).(ACA)_16_	256–288	F: ATTGAAAGTTGTATCGGTCGCTR: CTTGACATGAAAACGAGATCCA	MT074095	4	58	5′HEX
BLB2_om66	(AGT)_2_(AGC)_7_	337–345	F: CTATGGTCGTTTTCTCCGACATR: GACGTTTCTTCTCGGTTGTAGC	MT074097	8	60	5′HEX

**Table 4 insects-13-00543-t004:** Diversity indices of the five microsatellite loci developed for *Ootheca mutabilis*. Heterozygosity and polymorphic information content are *He* and PIC, respectively.

Locus Name	*He*	PIC
BLB_om1	0.75	75.0%
BLB_om17	0.65	58.8%
BLB_om32	0.80	78.5%
BLB_om33	0.84	83.1%
BLB_om66	0.59	50.1%
Average	0.73	69.1%

**Table 5 insects-13-00543-t005:** Gene diversity in the five populations of *Ootheca mutabilis*. Alphabet letters represent populations; (A) Northern moist farmlands, (B) Western mid-altitude farmlands, (C) Central wooden savannah, (D) Southern and Eastern Lake Kyoga basin and € North-western farmlands. Observed heterozygosity (*H_o_)*; expected heterozygosity (*H_e_*), and polymorphic information content (*PIC*) are shown.

Population Code	*H_o_*	*H_e_*	*PIC*
A	0.82	0.72	56.97%
B	0.80	0.68	51.48%
C	0.75	0.66	47.42%
D	0.84	0.73	57.72%
E	0.78	0.70	54.36%
Average	0.80	0.70	53.59%

**Table 6 insects-13-00543-t006:** Summary of genetic diversity indices for 658 bp fragment of the mtCOI partial gene. Number of haplotypes (h), Haplotype diversity (Hd), Average number of differences (K), and Nucleotide diversity (π) for each population are provided.

Population	Number of Sequences	Number of Segregating Sites	h	Hd	K	π
Dokolo	5	1	2	0.4	0.4	0
Lira	5	3	4	0.9	1.2	0
Oyam	6	2	3	0.6	0.67	0
Apac	5	1	2	0.4	0.4	0
Amuru	8	6	4	0.64	1.68	0
Gulu	4	1	2	0.5	0.5	0
Nwoya	7	4	3	0.52	1.14	0
Bulisa	10	4	4	0.53	0.96	0
Nakasongola	4	2	2	0.5	1	0
Amuria	9	2	3	0.56	0.61	0
Soroti	4	1	2	0.5	0.5	0
Adjumani	5	0	1	0	0	0
Zombo	3	1	2	0.67	0.67	0
Koboko	4	2	3	0.83	1	0
Moyo	3	2	2	0.67	1.33	0
Arua	4	2	2	0.5	1	0

**Table 7 insects-13-00543-t007:** (**a1**): Non-hierarchical AMOVA for all the sixteen populations of *O. mutabilis*, and (**a2**) hierarchical AMOVA for all the sixteen districts grouped into five agro-ecological zones based on SSR data; (**b1**) non-hierarchical AMOVA for all sixteen populations of *O. mutabilis,* and (**b2**) hierarchical AMOVA for all the sixteen populations grouped into their relative agro-ecological zones based on mt*COI* data.

(a1)				
Source of Variation	Sum of Squares		Variance Components	Percentage Variation (%)
Among populations	27.24		0.00122 Va	0.07
Within populations	264.05		1.780 Vb	99.93
Total	291.29		1.790	100
**(a2)**				
**Source of Variation**	**Sum of Squares**		**Variance Components**	**Fixation Indices**
Among groups	10.08		0.03722 Va	*F*_ST_ = 0.00582, *p* = 0.00880
Among populations within groups	17.16		−0.027 Vb	*F*_SC_ = −0.01523, *p* = 0.42326
Within populations	264.05		1.785 Vc	*F*_CT_ = 0.02073, *p* = 0.00587
Total	291.29		1.795	
(**b1**)				
**Source of Variation**	**d.f**		**Sum of Squares**	**Variance Components**
Among populations	15		5.83	−0.00671 Va
Within populations	70		264.05	0.42450 Vb
Total	85		291.29	0.418
(**b2**)				
**Source of Variation**	**d.f**	**Sum of Squares**	**Variance Components**	**Fixation Indices**
Among groups	4	1.71	0.00448 Va	*F*_ST_ = −0.01333, *p* = 0.73900
Among populations within groups	11	4.12	−0.01007 Vb	*F*_SC_ = −0.02429, *p* = 0.76051
Within populations	70	29.72	0.42450 Vc	*F*_CT_ = 0.01070, *p* = 0.30108
Total	85	35.55	0.419	

**Table 8 insects-13-00543-t008:** (**a1**). Spatial analysis of molecular variance (SAMOVA) of SSR data and (**a2**) SAMOVA of mt*COI* data showing the genetic diversity partitions in the *O. mutabilis* populations. (**a3**) and (**a4**) Analysis of molecular variance (AMOVA) of the genetic structure as suggested by SAMOVA for SSR and mt*COI* data respectively.

(a1)				
Source of Variation	d.f		Sum of Squares	Variance Components
Among groups	1		57,895.35	4273.04972 Va
Among populations within groups	14		143,369.46	387.00479 Vb
Within populations	156		935,627.3	5997.61091 Vc
Total	171		1,136,892.11	10,657.67
*F*_CT_ = 0.40094, *p* = 0.05767				
(**a2**)				
**Source of Variation**	**d.f**		**Sum of Squares**	**Variance Components**
Among groups	1		0.9284.904	0.04245 Va
Among populations within groups	14		4.9	−0.01442 Vb
Within populations	70		29.72	0.42450 Vc
Total	85		35.55	0.45
*F*_CT_ = 0.09381, *p* = 0.05474				
(**a3**)				
**Source of Variation**	**Sum of Squares**		**Variance Components**	**Percentage Variation (%)**
Among populations	27.24		0.00122 Va	0.07
Within populations	264.05		1.78482 Vb	99.93
Total	291.29		1.79	100
*F*_ST_ = 0.00069				
(**a4**)				
**Source of Variation**	**d.f.**	**Sum of Squares**	**Variance Components**	**Percentage Variation (%)**
Among populations	15	5.83	−0.00671 Va	−1.61
Within populations	70	29.72	0.42450 Vb	101.61
Total	85	35.55	0.42	100
*F*_ST_ = −0.01605				

## Data Availability

The data presented in this study are available on request from the corresponding author.

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
