# Peer review of "Population Genetic Structure of the Bean Leaf Beetle Ootheca mutabilis (Coleoptera: Chrysomelidae) in Uganda"

_insects, 2022, doi:10.3390/insects13060543_

Round 1

Reviewer 1 Report

The manuscript introduces new markers for a  economically important beetle species impacting agriculture. The manuscript highlights investigations on genetic structure and differentiation for which the data are hardly conclusive because only a low amount of markers could be included. The Manuscript should rather focus more on the molecular resources that are provided. In particular the manuscript could include also the samples that had been determined to contain mitochondrial haplotypes considered as from another species.  

The species of Ootheca in Uganda seem to be very difficult to distinguish and also only recently described (ref 26). It is puzzling why the authors did not include O. proteus into the analysis of microsatellites. They used barcoding to determine samples and the ones that have a haplotype resembling O. proteus had been removed from analysis. With this the haplotype is used as a diagnostic tool which might be misleading for such closely related groups. In addition it would provide information about cross species amplification and reproductive isolation between the species and increase information content of the manuscript. Therefore all individuals should be included regardless the species assignment hypothesis according to the haplotype.

The number of successfully amplifying markers is very, very low; outgoing from 81 primer pairs to 5 suitable ones. This should be discussed. Probably successrate could have been increased by comparing primers during construction with the contigs? Or number of reads the primer sequence could fit? Had something like this been undertaken? The comparison by blast to the reads created should be standard and can be used as a hint

The dataset does not show any structure, which is expected considering the low amount of markers included. The conclusion that k=2 reflects the number of genetic clusters which are admixed is misleading. Without individuals that can be clearly assigned to one or the other cluster this conclusion should not be made. It can be discussed as one possibility next to lack of structure because high levels of geneflow and low level of differentiation. Fst can be highlighted in this discussion.

At line 349 as well as the discussion covers Fst as low but indicating differentiation. The calculation of fst is not shown in the method part as far as I could see. The differentiation should be indicated in pairwise comparisons between populations, which are not shown in the main body of the text. Fst values might be wrongly interpreted, this has to be clarified and included in the discussion.

The discussion is overemphasizing genetic structure and differentiation which is not justified considering the low amount of markers used. Considerations about interbreeding do not make sense: In large populations mating between relatives is very unlikely so “high level of interbreeding” does not make much sense. Migrants and geneflow also not suitable measures when populations are not differentiated. The discussion should focus on the lack of differentiation and the likelihood of large population size and migration as main driver should be more clearly in the foreground. For example 427 the expression “the low but significant Fst” is unclear because the base is not given. Fst should be pairwise between populations, Fst between genetic clusters estimated by structure can be discussed but with caution.

Because these point have to be considered before a re-review, I am giving only a few minor suggestions for improvement at this point and not a detailed list. Generally the authors should also try to improve consistency between methods and results (e.g Fst).

Simple summary and Abstract are very similar. The simple summary should be simplified.

The text is subdivided with headers for very small paragraphs. Some should be condensed.

Line 196, the paragraph is formatted italic, probably accidently as a header

202, “primers were designed for all …” is probably wrong

Author Response

Response to Reviewer 1 Comments

Point 1: The manuscript introduces new markers for a  economically important beetle species impacting agriculture. The manuscript highlights investigations on genetic structure and differentiation for which the data are hardly conclusive because only a low amount of markers could be included. The Manuscript should rather focus more on the molecular resources that are provided. In particular the manuscript could include also the samples that had been determined to contain mitochondrial haplotypes considered as from another species.

Response 1: Thank you for this observation. We developed five microsatellite markers which we combined with mitochondrial COI partial gene. It is true the SSR markers are few. We have also included this limitation in our write up “Based on limited nuclear markers and the maternally inherited mtCOI marker, our findings showed that the two ancestral O. mutabilis genetic clusters could represent populations in the early stages of speciation. To gain a better knowledge of the evolutionary genetics and landscape adaptability of this major agricultural coleopteran pest complex, an in-depth population structure study based on whole genome sequencing would be necessary for O. mutabilis.”

We did not include other species that were not O. muatabilis however it would have been good if we had included them like you have advised. But we have included it in the manuscript that future research should consider testing these markers on them and other Ootheca species. We are currently unable to include the microsatellite data in our manuscript.

Point 2: The species of Ootheca in Uganda seem to be very difficult to distinguish and also only recently described (ref 26). It is puzzling why the authors did not include O. proteus into the analysis of microsatellites. They used barcoding to determine samples and the ones that have a haplotype resembling O. proteus had been removed from analysis. With this the haplotype is used as a diagnostic tool which might be misleading for such closely related groups. In addition it would provide information about cross species amplification and reproductive isolation between the species and increase information content of the manuscript. Therefore all individuals should be included regardless the species assignment hypothesis according to the haplotype.

Response 2: Thank you for this comment. It is true, Ootheca species are difficult to distinguish by applying only morphological analysis but by using molecular means via DNA barcoding, we were able to distinguish them. Analysing O. proteus with SSR markers now remains to be done for the future research because as of now we are really unable to include it in this manuscript.

Point 3: The number of successfully amplifying markers is very, very low; outgoing from 81 primer pairs to 5 suitable ones. This should be discussed. Probably successrate could have been increased by comparing primers during construction with the contigs? Or number of reads the primer sequence could fit? Had something like this been undertaken? The comparison by blast to the reads created should be standard and can be used as a hint

Response 3: Thank you for this comment. We reported a success rate of 6%. We checked the primers in the contigs during development. We have already submitted the sequences of the developed loci to NCBI.

Point 4:

The dataset does not show any structure, which is expected considering the low amount of markers included. The conclusion that k=2 reflects the number of genetic clusters which are admixed is misleading. Without individuals that can be clearly assigned to one or the other cluster this conclusion should not be made. It can be discussed as one possibility next to lack of structure because high levels of geneflow and low level of differentiation. Fst can be highlighted in this discussion.

Response 4: Thank you for this comment. We have indicated in the discussion that This study detected high levels of gene flow among populations of O. mutabilis and low genetic differentiation in the sampled agro-ecological zones, with the Ugandan population likely representing a single panmictic population with genetic contributions from two ancestral genetic clusters.

Point 5: At line 349 as well as the discussion covers Fst as low but indicating differentiation. The calculation of fst is not shown in the method part as far as I could see. The differentiation should be indicated in pairwise comparisons between populations, which are not shown in the main body of the text. Fst values might be wrongly interpreted, this has to be clarified and included in the discussion.

Response 5:  Thank you. We have included the equation of calculating the Fst as you advised i.e. Fst = (Ht − HS)/Ht, where Ht is the proportion of the heterozygotes in the total population and Hs the average proportion of heterozygotes in subpopulations

Point 6: The discussion is overemphasizing genetic structure and differentiation which is not justified considering the low amount of markers used. Considerations about interbreeding do not make sense: In large populations mating between relatives is very unlikely so “high level of interbreeding” does not make much sense. Migrants and geneflow also not suitable measures when populations are not differentiated. The discussion should focus on the lack of differentiation and the likelihood of large population size and migration as main driver should be more clearly in the foreground. For example 427 the expression “the low but significant Fst” is unclear because the base is not given. Fst should be pairwise between populations, Fst between genetic clusters estimated by structure can be discussed but with caution.

Response 6: Thank you for your comment. The high level of interbreeding, we put it in the context of interbreeding between individuals from different populations/locations. Also as explained in response 5, we have included the equation for calculating Fst.

Point 7: Because these point have to be considered before a re-review, I am giving only a few minor suggestions for improvement at this point and not a detailed list. Generally the authors should also try to improve consistency between methods and results (e.g Fst).

Response 7: Thank you for this observation about the inconsistancy between methods and results. We have made re-organisations in these sections by merging some short paragraphs and relocating some sections for example haplotype analysis has been relocated and merged with genetic structure subheading.

Point 8: Simple summary and Abstract are very similar. The simple summary should be simplified. The text is subdivided with headers for very small paragraphs. Some should be condensed. Line 196, the paragraph is formatted italic, probably accidently as a header. 202, “primers were designed for all …” is probably wrong

Response 8: Thank you for these critical observations. We have revised the simple summary and we have made it shorter as you advised. We have also merged some the short paragraphs and reduced the subheadings as you advised. The paragraph formatted italic, it was accidentary but it has been corrected.

We have also improved all the sections following all your comments including the first comments where you advised for example improving the introduction, research design, the methods, results, and the conclusions.

We are really very grateful for your comments and they have really helped us to improve our manuscript. Thank you very much.

Reviewer 2 Report

First and foremost: This is a very interesting contribution from the point of view of the design of management strategies as well as an understanding of the demographic history of O. mutabilis . The extent of the genomic analysis is impressive and the analysis is appropriate and relevant

A few issues that can help in the interpretation of the data and might provide more context for the reader

  1. Intro: Could the authors expand a bit more on the history of O.mutabilis in the area, when was it first reported as a pest, where was that.  What subfamily is it in, do the larval and adult feeding strategies match those of the subfamily?
  2. Materials and methods: Could the authors provide a more nuanced rationale for grouping the samples in five populations by agro-ecological zone despite originating from different districts. These groupings are not used throughout…, and the IDB is done with individual samples, what is the benefit of these groupings?
  3. Materials and methods: The genome sequencing to find microsatellites is impressive work, however it does not make it to the abstract or the intro. It would be best for the reader if it was better connected.
  4. Results: I am not sure we need the structure plots for all the K values used. The one that best fits the data is k=2 and that is the one of interest in my opinion.
  5. Discussion: Here is where having more context would be important. What has been found regarding population structure in other insect pests? Can the authors compare/ contrast with those studies the way they do with C. trifurcata?
  6. Discussion: Could the authors expand a bit more on the ideas of incipient speciation? As it is it is not a very convincing scenario for O.mutabilis; the degree of structure is low and the gene flow is high, which argues against that incipient speciation idea.

Author Response

Response to Reviewer 2 Comments

Point 1: Intro: Could the authors expand a bit more on the history of O. mutabilis in the area, when was it first reported as a pest, where was that.  What subfamily is it in, do the larval and adult feeding strategies match those of the subfamily?

Response 1: First and foremost we thank you for commending our work. We are really very grateful.

Changes: As you advised that we expand more on the history of O. mutabilis, we have added the information on when it was reported as a serious pest of beans in the first paragraph of the introduction.

Point 2: Materials and methods: Could the authors provide a more nuanced rationale for grouping the samples in five populations by agro-ecological zone despite originating from different districts. These groupings are not used throughout…, and the IDB is done with individual samples, what is the benefit of these groupings?

Response 2: Thank you for this observation.

Changes: The climatic conditions in the districts in an agro-ecological zone are mostly similar and therefore the climatic factors that may affect O. mutabilis in different districts in the same agro-ecological zone are the same.

We have now organised the manuscript and the population codes have been revised as they had not been presented well in the first submission. This information has been well presented in table 1.

We have expanded the analysis of IBD to make separate IBD for each population alone for both markers. The IBD values for each of the populations have beens well reported.

Also in some other sections, some figures have been relocated such as the haplotype network as one of the reviewers advised that expanded analysis be performed and reorganisation of some sections be done.

Point 3: Materials and methods: The genome sequencing to find microsatellites is impressive work, however it does not make it to the abstract or the intro. It would be best for the reader if it was better connected.

Response 3: We really appreciate you comments.

Changes: Here you required that we include information about next generation sequencing in the introduction. We have added the information on NGS to the introduction to include its advantages, attributes and its flexibility in relationtion microsatellite development.

Point 4: Results: I am not sure we need the structure plots for all the K values used. The one that best fits the data is k=2 and that is the one of interest in my opinion.

Response 4: Thank you for your critical observation on this point

Changes: We have made the changes as you required. From K3 to K5, they have been deleted and only K2 ha sremained as it is the one important being the true K. Earlier, we had not included the beyesian structuring basing on the mtCOI markers, but now we have included it and it is well presented where we also presented K2 as the most true K leaving other out as you advised.

Point 5: Discussion: Here is where having more context would be important. What has been found regarding population structure in other insect pests? Can the authors compare/ contrast with those studies the way they do with C. trifurcata?

Response 5: Thank you for this comment.

Changes: Ootheca species being a group of beetles whose genetic studies still have scarnty information, there was no beetle found whose behavior and characteristics resembles that of Ootheca mutabilis apart from Cerotoma trifurcata in USA that was reported to have similar feeding partterns as those of Ootheca species. By this we thought we would be speculating in case we compare other beetles which are not closely related with Ootheca species.

Point 6: Discussion: Could the authors expand a bit more on the ideas of incipient speciation? As it is it is not a very convincing scenario for O. mutabilis; the degree of structure is low and the gene flow is high, which argues against that incipient speciation idea.

Response 6: We appreciate you comments. Thank you for improving our write up.

Changes: The African rift valleys have been found to va caused incipient speciation in white fly (SSA1) as we wrote. Because of this senario we wrote that since we have not found the scenario of incipient speciation in this beetle, there would be a need to investigate whether those beetles that crossed the riftvalley may heve experienve the same scenario as the white fly.

We have also improved all  the sections following all your comments including the first comments where you advised for example improving the english, introduction, research design, the methods, results and the conclusions.

We are really very grateful for your comments and they have really helped us to improve our manuscript. Thank you very much.

Reviewer 3 Report

General comments

Kanyesigye et al. present a study on the population genetic structure of Ootheca mutabilis (Coleoptera: Chrysomelidae) in Uganda using five microsatellites and partial COI mt DNA sequences. The study of the genetic structure of O. mutabilis could be an interesting contribution; however, the way in which the work is presented and its unclear approach makes doubtful any contribution that could be derived from it. A key aspect is that the paper lacks hypotheses or questions to be tested. Without them, the paper is meaningless, nor can it be assessed how this information would contribute to the development of optimal management strategies for area-wide management of these destructive agricultural pests. The authors should focus on establishing an adequate theoretical framework for their work on the population genetic structure of this species. 

Introduction

The introduction should be completely restructured. Authors must establish questions, hypotheses or objectives related with study on population  genetic structure, based on a relevant theoretical framework. Everyone knows the importance of combining at least two types of molecular markers in this type of work and the goodness of microsatellites and mtDNA COI in this type of studies. it is not necessary to emphasize it. 

Methods

This section should be organized according to those questions posed in the introduction. Briefly state how the O. mutabilis specimens were identified and minimize how the microsatellites were obtained. In my opinion, the development of the microsatellites could be reported in another publication to evaluate widely the goodness of the strategy followed to obtain them.  The manner in which these two aspects are presented misses the purpose of the study and the analysis of the population genetic structure. In fact, both aspects only serve in this study to identify the specimens of O. mutabilis and to obtain the microsatellites for the study.

It should be specified, once the O. proteus specimens were identified, how many effective individuals of O. mutabilis were analyzed with both markers, to which samples of the districts they belong, as well as the number and name of the districts of each agro-ecological zone.  For each marker, the strategy and the analyses carried out with each of them should be specified. Briefly justify its application based on a hierarchical structure a priori AMOVA (established by the authors) or a posteriori BAPS OR STRUCTURE (delineated by the analysis methods themselves), clearly indicate the metrics to estimate the variation from each marker, estimate M instead of Nm, indicate the metrics of the degree of differentiation  and show the geographic distance between the samples of the districts into IBD.

Results

This section should be rewritten depending on the reanalysis strategy and metrics applied to each marker. 

Discussion

Finally, the authors should contrast the results of both markers, discuss them and project them in accordance with the proposed objective.

Figures and tables are pertinent, but headings and footnotes should be more generous and explicit. 

Author Response

Response to Reviewer 3 Comments

Point 1: Introduction: The introduction should be completely restructured. Authors must establish questions, hypotheses or objectives related with study on population  genetic structure, based on a relevant theoretical framework. Everyone knows the importance of combining at least two types of molecular markers in this type of work and the goodness of microsatellites and mtDNA COI in this type of studies. it is not necessary to emphasize it.

Response 1: Thank you very much. We found your comments very instrumental and helpful. Therefore, we have revised accordingly.

Changes: we have included more information concerning Ootheca mutabilis and why it was important to do this research on it. We have also included the purpose/objective of our study. One of the reviewers asked that we include a brief history of the pest and when it was reported in the area, more information on next generation sequencing, therefore, we have also included them. Also we have included concerning the goodness of combinig SSR and mtDNA markers since you advised that it is necessary us to include it.

As you required that english must be improved, we appreciate your critical observation. We have made revisions to english and we think that it is now better than it was before.

Point 2: Methods: This section should be organized according to those questions posed in the introduction. Briefly state how the O. mutabilis specimens were identified and minimize how the microsatellites were obtained. In my opinion, the development of the microsatellites could be reported in another publication to evaluate widely the goodness of the strategy followed to obtain them.  The manner in which these two aspects are presented misses the purpose of the study and the analysis of the population genetic structure. In fact, both aspects only serve in this study to identify the specimens of O. mutabilis and to obtain the microsatellites for the study.

It should be specified, once the O. proteus specimens were identified, how many effective individuals of O. mutabilis were analyzed with both markers, to which samples of the districts they belong, as well as the number and name of the districts of each agro-ecological zone.  For each marker, the strategy and the analyses carried out with each of them should be specified. Briefly justify its application based on a hierarchical structure a priori AMOVA (established by the authors) or a posteriori BAPS OR STRUCTURE (delineated by the analysis methods themselves), clearly indicate the metrics to estimate the variation from each marker, estimate M instead of Nm, indicate the metrics of the degree of differentiation  and show the geographic distance between the samples of the districts into IBD.

Response 2: Thank you. This was helpeful to our manuscript especially on re-organising and merging the sub sections in the methods to have a smooth flow.

Changes: we have re-organised the methods to ensure that the methods and their subsections are in chronological order rhyiming with the objective of the research. We have merged some subheadings for example; “Genome sequencing” subheading has been married with “quality check and raw read assembly”. Also, the sub heading “Microsatellite prediction,  primer design” has been combined with blast search of microsatellite sequences in GenBank” purposely to reduce the overexpression of the methods how we developed the microsatellites as you advised. Separating the manuscript to make two, one for microsatellite development and another for population genetic structure is a good thing like you advised but we considered putting them into one article, as it would help us to publish them as one body since these microsatellites were tested using the same samples that were used in the study. Also we have generally widened the coverage on the purpose of the study which is studying the population genetic structure of Ootheca mutabilis in uganda in all sections of the manuscript so that the tools used do not take the biggest share which may mask the purpose of the study.

We have clarified the number of O. mutabilis that were used in the study and the number of O. proteus that were removed from analysis after confirming their identity after DNA barcoding them. This information has been well presented than before in (table 1) as you advised. Also the agro-ecological zones and the districts per agro-ecological zone from which they were picked have been well presented. Also in table 1, we have made changes in population codes as it was not well presented in the first submission and it was kind of misleading, we have now made codes to be in order (A, B, C, D and E) than when they were before (A, D, C, B and E)

We carried out more analysis for both markers. We really appreciate your critical observation because in some cases some results had not been put in the first submission. We have now followed your directions and included the AMOVA tables for both markers as we had only included the AMOVA table for only SSR markers in the first submission. We have also included the bayesian population structure of both markers as only one for SSR analysis was presented in the first submission. In the first submission, we had presented five Ks (K2 (True value), K3, K4 and K5) but one of the reviewers advised that we only retain the K2 which is the true value and therefore, we have deleted other Ks as he advised.

As you advised that we make analysis for isolation by distance (IBD) per population instead of one for all the populations, that’s what we have done as you advised. We have analysed the samples for IBD per population for both markers and their values have been well included in the body of the manuscript well. Your observation was very instrumental to us on this analysis of IBD because in one of our converted GPS points, we found a minor error in one of the UTM coordinate points. This error entered was created after conversion from latitude longtude to UTM. Re-analysis has been a very good thing. I really thank you for your critical observation and for advising on reanalysis. In the current revised version the analysis is very fine and there is no more error.

Also we have relocated the haplotype network and brought it to the subheading of population genetic structure as where it was it was more isolated or mispalced.

Point 3: Results: This section should be rewritten depending on the reanalysis strategy and metrics applied to each marker. 

Response 3: We appreciate you for your advice on this section.

Changes: We have made re-organisations of some sub sections some of which were isolated and now there is a better flow. Since additional analysis was performed in some subsections, it is now rich with relavant analysis and a better order that before. In the legends of some figures and tables, population codes have been changed since the same codes originate from table 1 and since it was changes they were accordingly changed.

Point 4: Discussion: Finally, the authors should contrast the results of both markers, discuss them and project them in accordance with the proposed objective.

Response 4: We thank you again for you advice on this section.

Changes: Basing on the reanalysis and additional analysis in some result sections especially as describes in response number two, we did additional discussion for example on genetic variation, bayesian structuring and IBD so as to be inline with our study objective.

Point 5: Figures and tables are pertinent, but headings and footnotes should be more generous and explicit.

Response: Thank you for appreciating our figures and tables being pertnent. We are happy about that.

Changes: we have made changes to the table and figure legends to include more information sothat they bring out their purpose. This has been well explained in response number two.

We have also made changes to the conclusion. We have improved it to suit the purpose of the study.

We have also improved all  the sections following all your comments including the first comments where you advised for example improving the introduction, research design, the methods, results and the conclusions.

We are really very grateful for your comments and they have really helped us to improve our manuscript. Thank you very much.

Round 2

Reviewer 1 Report

The authors could not respond to the suggestion to include the data that were excluded for reasons (haplotype assignement to two different species) independent form the original question. The result of five markers that can be investigated that furthermore are not able to show structure of a populations should not be published more prominently then a technical note. I am not able to assess if this is reflected properly in the discussion, so that an exception can be made in this short time.

Author Response

Response to Reviewer 1 Comments

Point: The authors could not respond to the suggestion to include the data that were excluded for reasons (haplotype assignement to two different species) independent form the original question. The result of five markers that can be investigated that furthermore are not able to show structure of a populations should not be published more prominently then a technical note. I am not able to assess if this is reflected properly in the discussion, so that an exception can be made in this short time.

Response 1: Thank you for your comments. We have not refused to add the information that you suggested to us that we should add i.e. to add on the microsatellite markers and to include the O. proteus samples in the population genetic analysis. We are unable to provide this. We are very sorry, we wish we could be able but it is at this moment impossible for us.

We really appreciate your efforts rendered to improve our manuscript. Thank you very much.

Reviewer 3 Report

The authors missed the opportunity to make a thorough and detailed revision of their manuscript. In the introduction section, there are lines (lines 77-91) that are unnecessary and that if eliminated do not affect the purpose of the work. In addition, lines (113-116) should be moved to the material and methods section, specifically to the species identification part, since it is a justification of why to use the barcode.

The MM section should be restructured. It should describe how the collection was carried out, the number of specimens collected in each district and the agroecological zones to which the latter belong. The authors should make it clear that the insects were identified by morphological and molecular characteristics (barcode). And then establish that after combining both attributes, the number of specimens with which the genetic structure analysis was carried out was 87 individuals. Specimens of O. proteus should not be included in the genetic structure analysis. The pairwise distance table is a result, but it is unnecessary, since the reference sequences are included in the phylogeny. In any case placing the ranges of values in the phylogeny, and sending the phylogeny to supplementary material, because it has nothing to do with the genetic structure work.  Lines 131-138 should be deleted. they are repeated.  Statistical analysis to analyze whether or not there is genetic structure should be done by molecular marker, COI and microsatellites.  Start each of them with the diversity estimators. In the case of COI sequences, specify which haplotypes are found in each district and agroecological zones to appreciate the full extent of the haplotype network. Subsequently, perform the AMOVA establishing the hierarchy a priori: between agroecological zones, between districts within agroecological zones and finally within districts.  A posteriori analysis could also be performed, i.e. without establishing a priori groups, with SAMOVA or BAPS. From these analyses, authors would have paired Fst between agroecological zones and districts, of which they could estimate gene flow (Nm or M, I prefer M with migrate siftware). Use Fst across of relation Fst/1-Fst vs Ln geographic distance to establish if there is isolation by distance. Of course, other analyses can be done such as neutrality tests, population growth tests, etc. etc. etc. etc. In the case of microsatellites, similar analyses should follow along the same lines. It is recommended that the authors review the type of information provided by structure, since the way in which the results are described seems to confuse the interpretation.

The results section should be organized according to these analyses.

Finally, the discussion section should focus on the results obtained and not try to conform to a preconceived idea. From the data presented it is evident that there is no structure or rather there is a panmictic structure.  

Author Response

Response to Reviewer 3 Comments

Point 1: Introduction: The authors missed the opportunity to make a thorough and detailed revision of their manuscript. In the introduction section, there are lines (lines 77-91) that are unnecessary and that if eliminated do not affect the purpose of the work. In addition, lines (113-116) should be moved to the material and methods section, specifically to the species identification part, since it is a justification of why to use the barcode.

Response 1: Thank you very much for this comment. This has improved our manuscript.

Changes: We have deleted the information that was on lines 77-91 which you advised that it was unnecessary. We have also transffered the information about the mitochondrial markers which was on lines 113 to 116 when the reviewed copy returned.

Point 2: The MM section should be restructured. It should describe how the collection was carried out, the number of specimens collected in each district and the agroecological zones to which the latter belong. The authors should make it clear that the insects were identified by morphological and molecular characteristics (barcode). And then establish that after combining both attributes, the number of specimens with which the genetic structure analysis was carried out was 87 individuals. Specimens of O. proteus should not be included in the genetic structure analysis. The pairwise distance table is a result, but it is unnecessary, since the reference sequences are included in the phylogeny. In any case placing the ranges of values in the phylogeny, and sending the phylogeny to supplementary material, because it has nothing to do with the genetic structure work.  Lines 131-138 should be deleted. they are repeated.  Statistical analysis to analyze whether or not there is genetic structure should be done by molecular marker, COI and microsatellites.  Start each of them with the diversity estimators. In the case of COI sequences, specify which haplotypes are found in each district and agroecological zones to appreciate the full extent of the haplotype network. Subsequently, perform the AMOVA establishing the hierarchy a priori: between agroecological zones, between districts within agroecological zones and finally within districts.  A posteriori analysis could also be performed, i.e. without establishing a priori groups, with SAMOVA or BAPS. From these analyses, authors would have paired Fst between agroecological zones and districts, of which they could estimate gene flow (Nm or M, I prefer M with migrate siftware). Use Fst across of relation Fst/1-Fst vs Ln geographic distance to establish if there is isolation by distance. Of course, other analyses can be done such as neutrality tests, population growth tests, etc. etc. etc. etc. In the case of microsatellites, similar analyses should follow along the same lines. It is recommended that the authors review the type of information provided by structure, since the way in which the results are described seems to confuse the interpretation.

Response 2: We very much appreciate this landslide improvement and discipline you gave to our manuscript. It has been very instrumental and rewarding. Thank you very much.

Changes: We have revised the materials and methods section as you suggested to us. We have improved how the collection was done. For example, we have included “During sample collection, each garden from which samples were picked, a GPS coordinate was taken and each GPS point was regarded as one sample since only one sample (BLB) was considered for every GPS point where BLBs were recovered. From each farmers field where BLBs were found, at least one BLB was picked. In one of the districts (Lwengo) selected for sample collection in the Central wooden savannah agro-ecological zone, BLBs were not recovered at the time of collection (Table 1). This therefore reduced the number of districts as well as the number of samples in this agro-ecological zone. We have included table 1 which displays the Ugandan districts from which the samples were picked, GPS coordinates for each district and total number of samples picked per district.

We have also modified table two to include the numbers of leaf beetles considered per district after DNA barcoding. We included the numbers of samples which were found to be O. mutabilis and O. proteus and the districts they were picked. We have also included another district (Lwengo) which was considered during selection of districts for sample collection. Unfortunately, the beetle samples were not recovered from this district. Also after DNA barcoding of the samples, almost all the samples picked from Hoima district (Table 2) turned out to be O. proteus and only one sample was O.mutabilis. This prompted us to exclude this district as this would be a population with only one sample. This information was included in the write up.

Also, we have revised this manuscript and changed from considering agro-ecological zones as populations but instead considered districts as populations as you suggested. One of the reasons, some egro-ecological zones had bigger numbers of samples and many districts were involved in such agro-ecological zones. We also included that atleast every district from which samples were recovered, samples from that district were included in the study.

We explained well that the samples were selected based on morphological as well as DNA barcoding “As described by [26], the BLB samples studied for the population genetic structure were selected for DNA analysis based on color patterns of the elytra, head, thorax, abdomen, and legs. In this regard, we selected 99 samples based on their appearance as follows: M1 (O. mutabilis with elytra upper half black and lower half yellowish) (21 beetles), M2 (O. mutabilis with black elytra) (39 beetles), and M3 (O. mutabilis with brownish elytra) (39 beetles) (Fig. 2). This total included the samples found to be O. proteus before DNA barcoding as they could not be at all distinguished from O. mutabilis using only . All the BLBs with different colour appearances as explained above, and shown in Fig. 2 as well as reported by [26] were included in the analysis. This color distinction was made in order to examine potential genetic variations between O. mutabilis morphotypes. At least, every district from which BLB samples were recovered, was considered as well as some of the samples collected”.

As you advised, we did not include the O. proteus samples in the genetic analysis.

We have revised to exlude the pairwise distance table as you advised.

As you advised, we have deleted lines the repeated information that you advised ud to delete that was on lines 131-138 when the reviewed copy returned.

As advised, we have restructured the statistical analysis. We have provided the diversity estimators for both types of markers (mtCOI and microsatellites). We have included the analysis for both markers beginning with mtCOI. We have provided the haplotypes per agro-ecological zone, and per district as well. We have inferred the number of haplotypes, haplotype diversity, nucleotide diversity and the number of segregating sites for the total number of samples and as well for each district as you advised. We have also inferred the haplotype network indicating the haplotypes per district or population and those per sgro-ecological zone.

We have performed the analysis of molecular variance using Arlequin37 for the data of both types of markers. We have provided the results for when hierarchical AMOVA was performed and when non hierarchical AMOVA was performed.

As you advised we have performed the spatial analysis of molecular variance (SAMOVA) using SAMOVA2.0. with K set starting from K = 2 to K = 10. The structure suggested by SAMOVA2.0 for data from both types of markers were inferred using Arlequin35 to perform the AMOVA.

For all the analyses we performed, we modified the data analysis section to include all the details how the analysis was performed.

We also revised and modified the analysis by STRUCTURE. We have now included the district names so that the information provided is very clear.

We have not included the analysis from migrate-n software in this revised version. I had my machine get serious technical problems and it has really given me a little harder time. I am kindly with great respect requesting you that you allow so that I provide this result from migrate-n at a later date shortly after this re-submission, during the review process. It has taken some time and the submission date has overextended. Therefore I request that you consider this and I must submit to you the gene flow results analyzed using MIGRATE. Also, isolation by distance as you advisedUse Fst across of relation Fst/1-Fst vs Ln geographic distance to establish if there is isolation by distance, and neutrality tests”. I request to provide these results shortly after this submission of this revised version. I am fixing my machine as it is insufficient to perform these analyses without this fixing. I am very sure these results will be provided as I have promised. Thank you.

In the discussion section concerning structure, we have made it clear as you advised because it was confusing. We have restructure and put it clear as follows: “Our study areas were only limited to Uganda, however, BLBs have been reported to occur elsewhere for example in East African countries such as Tanzania and other countries in West Africa [26]. It would be a greater opportunity to understand their genetic status as the East African Rift Valleys have been shown to support population substructure and/or early speciation in both invertebrates (e.g., [11,47,28]) and vertebrates [13,20]. It remains to be investigated whether the geographical distribution of O. mutabilis, which crossed the Rift Valley, may have resulted in comparable incipient speciation as observed in the cassava whitefly Bemisia 'SSA1' species using a whole genome analysis technique [11]. We realized that this information was placed in the manuscript and it seemed to confuse as anyone would think that we found a substructure in our work. We were ideally concerned with the populations of O. mutabilis that may have crossed the rift valley which has been shown to cause substructure and it causes separation and populations may not be able to meet to exchange genes hence speciation.

Point: The results section should be organized according to these analyses.

Response: Thank you so much for you comments.

Changes: We have restructured the results section to include the results from the analyses performed for example, the analysis of molecular variance (AMOVA) analyzed using Arlequin. On this, we presented the results of hierarchical as well as non hierarchical AMOVA for both types of markers. We also included the results from SAMOVA2.0 for both types of markers. We included, the results from DnaSP6 and POPART concerning the results from haplotype analysis and haplotype network inferences. We have presented haplotype analysis per district and for all the samples. We have also presented the haplotype network showing which haplotypes that belong to every district. In the result section, we also included the presentation of structure showing the structure of samples per district.

 Point: Finally, the discussion section should focus on the results obtained and not try to conform to a preconceived idea. From the data presented it is evident that there is no structure or rather there is a panmictic structure. 

 Response: Thank you for this comment.

Changes: we have discussed the results that we have included to our manuscript. We discussed the AMOVA results and also the SAMOVA results. We also have discussed the haplotype analysis results as well as the results from the structure. Surely, your comments are very honest, our results show a panmictic structure and in our discussion we have included all this information.

We also revised the conclusion to include more information pertinent to the purpose of our research.

We really appreciate your comments, they have really guided us greatly. Therefore we are very grateful. Thank you very much.
